# An Unsupervised Information-Theoretic Perceptual Quality Metric

**Sangnie Bhardwaj**
Google Research
sangnie@google.com

**Ian Fischer**
Google Research
iansf@google.com

**Johannes Ballé**
Google Research
jballe@google.com

**Troy Chinen**
Google Research
tchinen@google.com

## Abstract

Tractable models of human perception have proved to be challenging to build. Hand-designed models such as MS-SSIM remain popular predictors of human image quality judgements due to their simplicity and speed. Recent modern deep learning approaches can perform better, but they rely on supervised data which can be costly to gather: large sets of class labels such as ImageNet, image quality ratings, or both. We combine recent advances in information-theoretic objective functions with a computational architecture informed by the physiology of the human visual system and unsupervised training on pairs of video frames, yielding our Perceptual Information Metric (PIM)[1]. We show that PIM is competitive with supervised metrics on the recent and challenging BAPPS image quality assessment dataset and outperforms them in predicting the ranking of image compression methods in CLIC 2020. We also perform qualitative experiments using the ImageNet-C dataset, and establish that PIM is robust with respect to architectural details.

## 1 Introduction

Many vision tasks require the assessment of subjective image quality for evaluation, including compression and restoration problems such as denoising, deblurring, colorization, etc. The success of many such techniques is measured in how similar the reconstructed image appears to human observers, compared to the often unobserved original image (the image before compression is applied, the actual scene luminances without the noise of the sensor, etc.). Predicting subjective image quality judgements is a difficult problem.

So far, the field has been dominated by simple models with few parameters that are hand-adjusted to correlate well with human mean opinion scores (MOS), such as SSIM and variants (Wang, Bovik, et al., 2004; Wang, Simoncelli, and Bovik, 2003). This class of models captures well-documented phenomena observed in visual psychology, such as spatial frequency dependent contrast sensitivity (Van Nes and Bouman, 1967), or are based on models of early sensory neurons, such as divisive normalization, which explains luminance and/or contrast adaptivity (Heeger, 1992). It is remarkable these models perform as well as they do given their simplicity. However, it is also clear that these models can be improved upon by modeling more complex and, at this point, potentially less well understood properties of human visual perception. Examples for this can be found in the recent literature (R. Zhang et al., 2018; Chinen et al., 2018). The underlying hypothesis in these models is that the same image features extracted for image classification are also useful for other tasks, including

prediction of human image quality judgements. However, this approach requires model fitting in several stages, as well as data collected from human raters for training the models. First, human responses are collected on a large-scale classification task, such as the ImageNet dataset (Russakovsky et al., 2015). Second, a classifier network is trained to predict human classifications. Third, more human responses are collected on an image quality assessment (IQA) task. Fourth, the features learned by the classifier network are frozen and augmented with additional processing stages that are fitted to predict the human IQA ratings. This process is cumbersome, and gathering human responses is costly and slow.

In this paper, we follow a different, and potentially complementary approach. Our work is inspired by two principles that have been hypothesized to shape sensory processing in biological systems. One goes back to as early as the 1950s: the idea of *efficient coding* (Attneave, 1954; Barlow, 1961). The efficient coding hypothesis proposes that the internal representation of images in the human visual system is optimized to efficiently represent the visual information processed by it. That is, the brain *compresses* visual information. The other principle is *slowness* (Földiák, 1991; Mitchison, 1991; Wiskott, 2003), which posits that image features that are not persistent across small time scales are likely to be uninformative for human decision making. For example, two images taken of the same scene within a short time interval would in most cases differ in small ways (e.g., by small object movements, different instances of sensor noise, small changes in lighting), but informative features, such as object identity, would persist. The concept of slowness is related to the information theoretic quantity of *predictive information* (Bialek et al., 2001), which we define here as the mutual information between the two images (Creutzig and Sprekeler, 2008). In this view, the information that is not predictive is likely not perceived, or at least does not significantly contribute to perceptual decision making.

We conjecture that by constructing a metric based on an image representation that efficiently encodes temporally persistent visual information we will be able to make better predictions about human visual perception. We find that such a metric, PIM, is competitive with the fully supervised LPIPS model (R. Zhang et al., 2018) on the triplet human rating dataset published in the same paper (BAPPS-2AFC), and outperforms the same model on the corresponding just-noticeable-difference dataset (BAPPS-JND). Remarkably, our model achieves this without requiring any responses collected from humans whatsoever.

# 2 Perceptual Information Metric

A principled approach to defining an IQA metric is to construct an image representation, for example, by transforming the image via a deterministic function into an alternate space, and then measuring distances in this space. Thresholding the distance between two image representations can be used to make predictions about the just-noticeable difference (JND) of image distortions; comparing the distance between two alternate image representations and a reference representation can be used to make predictions about which of the alternate images appears more similar to the reference.

More generally, we can construct a probabilistic representation by representing an image as a probability distribution over a latent space. This gives the model flexibility to express uncertainty, such as when image content is ambiguous. Here, we construct a multi-scale probabilistic representation $q(Z|X)$, where $X$ is an image, and $Z$ is a random variable defined on a collection of multi-scale tensors. The encoder distribution $q$ is parameterized by artificial neural networks (ANNs) taking images as inputs (Figs. 1 and 2), which allows amortized inference (i.e., the latent-space distribution can be computed in constant time). To use this representation as an IQA metric, we measure symmetrized Kullback–Leibler divergences between $q$ given different images, which gives rise to our Perceptual Information Metric (PIM).

To train this representation unsupervisedly, we are guided by a number of inductive biases, described in the following sections. Note that implementation details regarding the optimization (e.g. optimization algorithm, learning rate) and pre-processing of the training dataset can be found in the appendix.

## 2.1 Choice of objective function

Our choice of objective function is informed by both the efficient coding and the slowness principles: it must be compressive, and it must capture temporally persistent information. We choose a stochastic

variational bound on the Multivariate Mutual Information (MMI) to satisfy these constraints, called IXYZ. IXYZ learns a stochastic representation, $Z$, of two observed variables, $X$ and $Y$ (in this case two temporally close images of the same scene), such that $Z$ maximizes a lower bound on the Multivariate Mutual Information (MMI) $I(X;Y;Z)$ (Fischer, 2019a):

$$
\begin{aligned}
I(X;Y;Z) &= \int dx\, dy\, p(x,y) \int dz\, p(z|x,y) \log \frac{p(z|x)p(z|y)}{p(z)p(z|x,y)} \\
&= \mathbb{E} \log \frac{p(z|x)p(z|y)}{p(z)p(z|x,y)} \geq \mathbb{E} \log \frac{q(z|x)q(z|y)}{\hat{p}(z)p(z|x,y)} \equiv \text{IXYZ} \quad (1)
\end{aligned}
$$

Here, $p(z|x,y)$ is our *full encoder* of the pair of images, and $q(z|x)$ and $q(z|y)$ are variational approximations to the encoders of $X$ and $Y$, which we call *marginal encoders*, since they must learn to marginalize out the missing conditioning variable (e.g., $y$ is the marginalization variable for $q(z|x)$). All three of those can be parameterized by the outputs of neural networks. $\hat{p}(z)$ is a minibatch marginalization of $p(z|x,y)$: $\hat{p}(z) \equiv \frac{1}{K} \sum_{i=1}^{K} p(z|x_i, y_i)$, where $K$ is the number of examples in the minibatch.[2] These substitutions make it feasible to maximize IXYZ using stochastic gradient descent. To see that this objective is compressive, decompose the MMI into three additive terms, which are each lower bounded by three terms that analogously constitute IXYZ:

$$
I(X;Y;Z) = \left\{
\begin{array}{rcccc}
I(Z;X,Y) &=& \mathbb{E} \log \frac{p(z|x,y)}{p(z)} &\geq& \mathbb{E} \log \frac{p(z|x,y)}{\hat{p}(z)} \\
-I(X;Z|Y) &=& -\mathbb{E} \log \frac{p(z|x,y)}{p(z|y)} &\geq& -\mathbb{E} \log \frac{p(z|x,y)}{q(z|y)} \\
-I(Y;Z|X) &=& -\mathbb{E} \log \frac{p(z|x,y)}{p(z|x)} &\geq& -\mathbb{E} \log \frac{p(z|x,y)}{q(z|x)}
\end{array}
\right\} = \text{IXYZ} \quad (2)
$$

By maximizing IXYZ, we maximize a lower bound on $I(Z;X,Y)$, encouraging the representation $Z$ to encode information about $X$ and $Y$. Simultaneously, we *minimize* upper bounds on $I(X;Z|Y)$ and $I(Y;Z|X)$. This discourages $Z$ from encoding information about $X$ that is irrelevant for predicting $Y$, and vice versa.

## 2.2 Parameterization of encoder distributions

As stated above, we seek a distribution over the representation conditioned on an image; the marginal encoder $q(z|x)$ is the core piece we are trying to learn. The full encoder $p(z|x,y)$ is only necessary for training, and it is the only distribution we need to sample from. We are able to use mixture distributions for the marginal encoders, since taking gradients of their log probability densities is tractable. This is a benefit of IXYZ compared to other approaches like the Variational Information Bottleneck (VIB) (Alemi et al., 2017) or the Conditional Entropy Bottleneck (Fischer, 2019b).[3]

Using mixture distributions for $q(z|x)$ allows us to learn very expressive inference encoders that, in the limit of infinite mixtures, can exactly marginalize the full encoder distribution, $p(z|x,y)$. In practice, we find that a mixture of 5 Gaussians for $q(z|x)$ and $q(z|y)$ is sufficient to learn powerful models with tight upper bounds on the compression terms. For the full encoder, we parameterize only the mean of a multivariate Gaussian, setting the variance to one. Correspondingly, the marginal encoders are mixtures of multivariate Gaussians, also with unit variance for each mixture component.

We evaluate PIM by estimating the symmetrized Kullback–Leibler divergence using Monte Carlo sampling:

$$
\begin{aligned}
\text{PIM}(\boldsymbol{x}, \boldsymbol{y}) &= D_{\text{KL}}[q(\boldsymbol{z}|\boldsymbol{x}) \parallel q(\boldsymbol{z}|\boldsymbol{y})] + D_{\text{KL}}[q(\boldsymbol{z}|\boldsymbol{y}) \parallel q(\boldsymbol{z}|\boldsymbol{x})] \\
&\approx \frac{1}{N} \sum_{\boldsymbol{z}_n \sim q(\boldsymbol{z}|\boldsymbol{x})}^{N} \log \frac{q(\boldsymbol{z}_n|\boldsymbol{x})}{q(\boldsymbol{z}_n|\boldsymbol{y})} + \frac{1}{N} \sum_{\boldsymbol{z}_n \sim q(\boldsymbol{z}|\boldsymbol{y})}^{N} \log \frac{q(\boldsymbol{z}_n|\boldsymbol{y})}{q(\boldsymbol{z}_n|\boldsymbol{x})}. \quad (3)
\end{aligned}
$$

While this yields accurate results, it is not an expression that is differentiable with respect to the images, which would be useful for optimization purposes. We note, however, that the predictive performance of PIM suffers only slightly when collapsing the marginal encoders to just one mixture

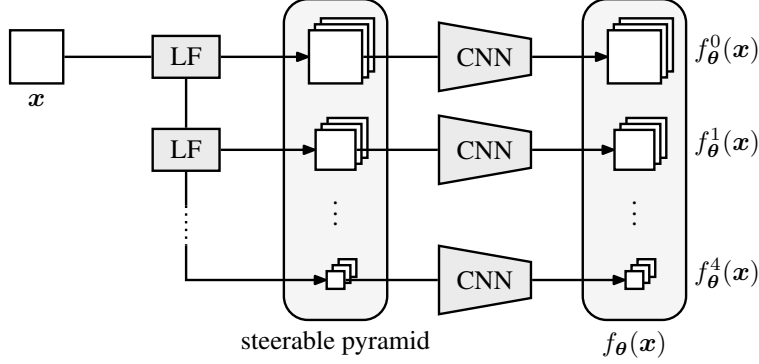

Figure 1: System diagram of frontend shared between all encoders. An image ($\boldsymbol{x}$) is decomposed using linear filtering (LF) into a multi-scale pyramid representation. Each scale is then subjected to processing by a convolutional neural network (CNN) with trained parameters $\boldsymbol{\theta}$, which are not shared across scales. The result is a multi-scale collection of tensors we denote $f_{\boldsymbol{\theta}}(\boldsymbol{x})$.

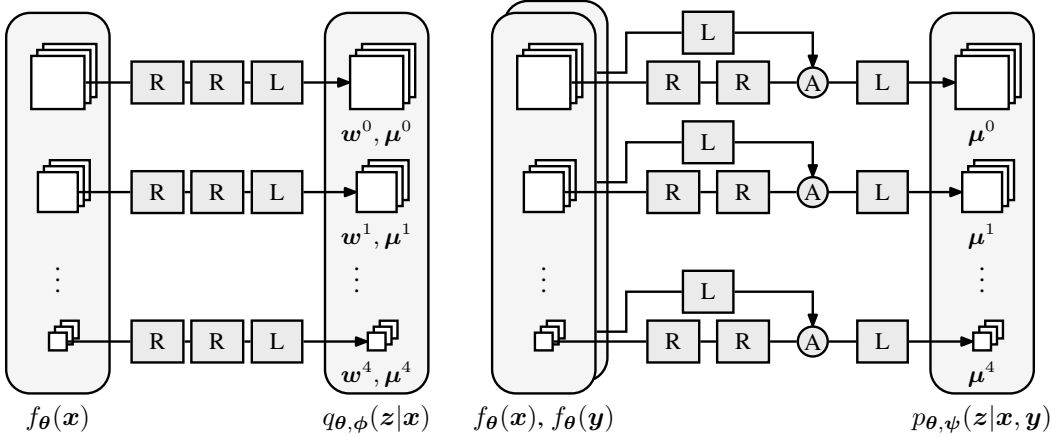

Figure 2: **Left:** system diagram of marginal encoder $q_{\boldsymbol{\theta},\phi}(\boldsymbol{z}|\boldsymbol{x})$ (identical for $q_{\boldsymbol{\theta},\phi}(\boldsymbol{z}|\boldsymbol{y})$). Each tensor produced by the frontend is fed into a three-layer neural network consisting of two rectified linear layers (R) with 50 units each, and one linear layer (L), which outputs 5 mixture weights and 5 mean vectors of length 10 per spatial location and scale. The parameters of both marginal encoders are identical, not shared across scales, and denoted together as $\phi$. **Right:** system diagram of full encoder $p_{\boldsymbol{\theta},\psi}(\boldsymbol{z}|\boldsymbol{x},\boldsymbol{y})$, whose parameters we denote as $\psi$. Each tensor of $f_{\boldsymbol{\theta}}(\boldsymbol{x})$ is fed into a three-layer network with 10 units per layer, the final outputs representing one mean vector per spatial location and scale. $f_{\boldsymbol{\theta}}(\boldsymbol{y})$ is subjected to a linear layer, which outputs 10 factors and 10 offsets of an additional elementwise affine transformation (A) of the activations of the second layer (R). All layers are separable in space, i.e. convolutions with $1 \times 1$ spatial support.

component (Table 1). In that case, the KL divergence collapses to a simple squared distance between the means, which is desirable both in terms of computational complexity and interpretability – we treat the latent space then as a *perceptually uniform space* like, for example, the $\Delta E^*$ metric developed by the International Commission on Illumination (CIE) for color perception. In the remainder of the paper, we report KL divergences as per Eq. (3).

## 2.3 Choice of dataset

In line with the slowness principle, we seek to extract the predictive information of two images $X$ and $Y$ taken within short time intervals. We approximate this by extracting pairs of consecutive frames from video sequences, 1/30th of a second apart, which is generally short enough to simulate continuous motion to the human visual system (HVS). The nature of the imagery used for training can of course vary according to video content. For example, the statistics of horizontal and vertical edges, as well as the amount of and type of texture tends to differ between motifs, e.g., in city scenes vs.

nature scenes. Importantly, image composition can have an effect on the distribution of object scales. Natural visual environments tend to exhibit *scale equivariance*: both the statistics of images recorded by photographers as well as the properties of feature detectors in the early human visual system themselves are self-similar across scales, such as the neurons found in cortical region V1 (Field, 1987; Ruderman, 1997). Our training data is largely determined by its availability: we leveraged the database of publicly available videos on YouTube. In our experiments, we were forced to reduce the spatial resolution of the video to eliminate existing compression artifacts, which limits the expression of scale equivariance in the training dataset. We addressed this issue by imposing approximate scale equivariance on the computational architecture described in the next section.

## 2.4 Choice of architectural components

As we specify the model details, we now indicate learned parameters with Greek subscripts, and vectors with bold. The mean parameters of the multivariate Gaussians and the mixture weights of the marginal encoders are computed by a multi-stage architecture of ANNs, as shown in Figs. 1 and 2. All encoder distributions are defined jointly on a collection of multi-scale latent variables (i.e., $z = \{z^0, \ldots, z^4\}$), but the distribution parameters (means, and mixture weights for the marginal encoders) are computed independently for each scale. Each $z^s$ has 10 dimensions per spatial location.

All three encoders share the same computational frontend with parameters $\boldsymbol{\theta}$, which we denote $f_{\boldsymbol{\theta}}(\boldsymbol{x})$. This frontend, shown in Fig. 1, consists of a multi-scale transform with no trainable parameters, followed by a set of convolutional neural networks (CNNs). We use a steerable pyramid (Simoncelli and Freeman, 1995) with 3 bandpass scales. Each bandpass scale has 2 oriented subbands, making for 8 sub-bands per color channel total, including the highpass and lowpass residuals. Note that the spatial resolution of the scales varies as defined by the multi-scale pyramid. The CNNs each consist of 4 layers with 64, 64, 64 and 3 units, respectively, a spatial support of $5 \times 5$, and ReLU activation function. The marginal encoder takes the output of the frontend as its input, and outputs the conditional mean vectors and weights of a 10-dimensional Gaussian mixture distribution with 5 unit-variance mixture components (Fig. 2, left panel) per spatial location and scale. The marginal encoders for $X$ and $Y$ are identical, i.e. share all parameters. The full encoder takes the output of the frontend to both $\boldsymbol{x}$ and $\boldsymbol{y}$, and computes the conditional mean via a separate set of neural networks (Fig. 2, right panel).

The architectural inductive bias can thus be summarized in three constraints: *spatial translation equivariance* via the convolutionality of all components of the architecture; *approximate spatial scale equivariance*, by separating the image representation into multiple scales and forcing the computation to be independent across scales (effectively forcing the model to equally weight information from each of the scales, although it does not explicitly assume scale equivariance of the computation); and *temporal translation equivariance* by sharing parameters between the marginal encoders.

## 3 Evaluation

We assess the properties of our unsupervised representation in four ways. First, we use PIM to make predictions on two datasets of human image quality ratings, previously collected under a two-alternative forced choice (2AFC) paradigm, comparing it to other recently proposed metrics on this task. Second, we posit that shifting an image by a small number of pixels should only have negligible effect on human perceptual decision making. We quantify this on PIM and a variety of metrics. Third, we generalize this experiment to gather intuitions about the relative weighting of different types of distortions via the ImageNet-C dataset (Hendrycks and Dietterich, 2019). Finally, we assess the robustness of our approach using a number of ablation experiments.

### 3.1 Predictive performance on BAPPS/CLIC 2020

We evaluate the performance of PIM on BAPPS, a dataset of human perceptual similarity judgements (R. Zhang et al., 2018). BAPPS contains two task-specific datasets: a triplet dataset, in which humans were asked to identify the more similar of two distorted images compared to a reference image, named "2AFC", and a dataset of image pairs judged as identical or not, named "JND". For reporting the performance of PIM, we follow the original authors: for BAPPS-2AFC, we report the fraction of human raters agreeing with the metric as a percentage. For BAPPS-JND, we report mAP, an

| Metric | BAPPS-2AFC | | | | | | | BAPPS-JND | | |
|---|---|---|---|---|---|---|---|---|---|---|
| | Trad. | CNN | SuperRes | Deblur | Coloriz. | Interp. | All | Trad. | CNN | All |
| MS-SSIM | 61.24 | 79.34 | 64.62 | 58.88 | 57.28 | 57.29 | 63.26 | 36.20 | 63.77 | 52.50 |
| NLPD | 58.23 | 80.29 | 65.54 | 58.83 | 60.07 | 55.40 | 63.50 | 34.91 | 61.49 | 50.80 |
| LPIPS Alex | 71.91 | **83.55** | **71.57** | 60.45 | **64.94** | 62.74 | 68.98 | 46.88 | 67.86 | 59.47 |
| LPIPS Alex-lin | 75.27 | **83.52** | 71.11 | 61.17 | **65.17** | **63.35** | **69.53** | 51.92 | 67.78 | 61.50 |
| PIM | **75.74** | 82.66 | 70.33 | **61.56** | 63.32 | 62.64 | **69.11** | **60.07** | **68.49** | 64.40 |
| PIM-1 | **76.41** | 82.81 | 69.14 | **61.53** | 63.40 | 62.48 | 68.82 | **60.00** | **68.69** | **64.46** |

Table 1: Scores on BAPPS. Best values are underlined, the bold values are within 0.5% of the best. All numbers reported for LPIPS were computed using the code and weights provided by R. Zhang et al. (2018). Categories follow the same publication and "all" indicates overall scores.

| Metric | Spearman's $\rho$ |
|---|---|
| PSNR | –0.139 |
| MS-SSIM (Wang, Simoncelli, and Bovik, 2003) | 0.212 |
| SSIMULACRA (Sneyers, 2017) | –0.029 |
| *Butteraugli* (2019) (1-norm) | –0.461 |
| *Butteraugli* (2019) (6-norm) | –0.676 |
| LPIPS Alex-lin (R. Zhang et al., 2018) | –0.847 |
| **PIM** | **–0.864** |

Table 2: Rank correlations of metrics on CLIC 2020 human ratings. Ideally, correlation values are either 1 or -1. According to their definitions, PSNR, MS-SSIM, and SSIMULACRA are expected to be positively correlated. The others are expected to be inversely correlated with the human ratings.

area-under-the-curve score.

We compare PIM to two traditional perceptual quality metrics, MS-SSIM and NLPD, and the more recent LPIPS, published with the BAPPS dataset (Table 1). For LPIPS, we use two models provided by the authors: LPIPS Alex, which uses pretrained AlexNet features, and LPIPS Alex-lin, which adds a linear network on top, supervisedly tuned on the 2AFC dataset. The numbers reported for PIM are the best out of 5 repeats. PIM scores $69.06 \pm 0.07\%$ on 2AFC and $64.14 \pm 0.15\%$ on BAPPS-JND on average across the 5 runs. The best model at 64.40% performs the best out of all metrics on BAPPS-JND, outperforming both LPIPS Alex and Alex-lin. On BAPPS-2AFC, it scores 69.06%, outperforming LPIPS Alex and performing at about the same level as Alex-lin. The single-component model, for which the KL divergence in Eq. (3) collapses to a Euclidean distance, performs only slightly worse on 2AFC at 68.82%.

R. Zhang et al. (2018) report that on the BAPPS-2AFC dataset, the cross-rater agreement is 73.9%. As such, it is remarkable that PIM performs this well, absent any training data that involves humans (neither ImageNet classification labels nor quality ratings, as used in LPIPS). Our results can thus be interpreted as empirical evidence supporting the validity of the inductive biases we employ as characterizations of the human visual system.

To assess the utility of PIM in predicting human ratings on larger images, we used it to predict the human ranking of contestants in the Workshop and Challenge on Learned Image Compression (CLIC 2020). The image compression methods were ranked using human ratings and the ELO system (Elo, 2008). Subsequently, the same patches viewed by the raters were evaluated using each metric, and the Spearman rank correlation coefficient was computed between the ranking according to each metric and the ranking according to ELO. Table 2 shows that PIM performs best in predicting the ranking of learned compression methods.

## 3.2 Invariance under pixel shifts

Metrics such as MS-SSIM and NLPD, which operate on pairwise comparisons between corresponding spatial locations in images, typically do not perform well on geometric transformations (and older benchmarks often do not contain this type of distortion). More complex metrics such as LPIPS and PIM should perform better. To verify this, we follow the approach of Ding et al. (2020) and shift the

| Metric \ Shift | BAPPS-2AFC | | | | | BAPPS-JND | | | | |
|---|---|---|---|---|---|---|---|---|---|---|
| | 1 | 2 | 3 | 4 | 5 | 1 | 2 | 3 | 4 | 5 |
| MS-SSIM | −1.18 | −7.62 | −11.10 | −12.70 | −13.50 | −11.60 | −17.60 | −19.98 | −21.20 | −21.80 |
| NLPD | −2.18 | −7.22 | −10.40 | −12.40 | −13.80 | −13.20 | −18.10 | −19.99 | −20.70 | −21.00 |
| LPIPS Alex | −0.06 | −0.25 | −0.34 | −0.48 | −0.68 | −3.90 | −6.96 | −8.64 | −9.77 | −11.50 |
| LPIPS Alex-lin | −0.11 | −0.18 | −0.27 | −0.30 | −0.48 | −1.99 | −3.34 | −4.56 | −5.49 | −7.06 |
| PIM | **−0.03** | **−0.07** | **−0.13** | **−0.27** | **−0.40** | **−0.1** | **−0.97** | **−2.61** | **−4.52** | **−6.33** |

Table 3: Score differences on pixel-shifted BAPPS. Bold values indicate best for a column.

reference images in BAPPS by a few pixels ($\leq 5$ for a 64x64 image). Because the shifts are very small, we reasonably assume that human judgements of the modified pairs would be essentially unchanged.

The score differences w.r.t. the unmodified BAPPS results on this transformed dataset are presented in Table 3. MS-SSIM and NLPD lose over 7 percentage points on BAPPS-2AFC when shifting by only 2 pixels, while the deep metrics (including PIM) have only a negligible decrease. On BAPPS-JND the effect is even more stark, where both traditional metrics' scores decrease by almost 12 when shifting by 1 pixel, and 18 for 2 pixels. LPIPS' scores show a more noticeable decrease on shifting by 2 pixels as well, 7 for LPIPS Alex and 3 for LPIPS Alex-lin. PIM performance decreases by less than one.

### 3.3 Qualitative comparisons via ImageNet-C

Hendrycks and Dietterich (2019) provide a dataset, ImageNet-C, which consists of 15 corruption types with 5 different severity levels applied to the ImageNet validation set, meant as a way to assess classifiers. We compared the predictions PIM, LPIPS and MS-SSIM make with respect to the various types of corruptions subjectively; we also added two additional corruptions, pixel shift and zoom, also with 5 levels of severity. We observed two significant effects across the dataset: First, MS-SSIM deviates from the other metrics on geometric distortions such as zoom, shift, and "elastic transform", further giving weight to the observations made in the previous section. Second, we noted that LPIPS deviates from PIM and MS-SSIM on global changes of contrast and similar distortions such as "fog". Subjectively, we found that LPIPS was not sensitive enough to this kind of corruption. We speculate this may be due to LPIPS using pre-trained features from a classifier network, which in principle should be invariant to global changes in lighting. Indeed, it is plausible that the invariances of a model predicting image quality should not necessarily be identical to the invariances of a network solving a classification task.

To quantify these effects, we conducted the following simple experiment: For a given metric, we computed the metric value between a reference and a corrupted image and then found an equivalent amount of Gaussian noise to add to the reference that yields the same metric value. We plotted the average standard deviation of required Gaussian noise across the dataset in Fig. 3(a). Clearly, MS-SSIM is more sensitive to the "zoom" corruption than the other metrics, and LPIPS is less sensitive to "fog" than MS-SSIM and PIM, both relative to their sensitivity to Gaussian noise. Fig. 3(b) and (c) show the corresponding noise strengths for a representative image. It is interesting to note that the images corrupted by fog are distinguishable at a glance, even at the lowest corruption strength. However, they remain recognizable (i.e. classifiable), suggesting that a classifier, which LPIPS is based on, should be invariant to the corruption.

Further details of this experiment can be found in the appendix. Note that these results are not necessarily conclusive, and further experiments are needed to assess the different classes of metrics, such as with the methods presented by Wang and Simoncelli (2008) or Berardino et al. (2017).

### 3.4 Ablations

To study the effect of the different inductive biases we impose, specifically the loss, the use of a multi-scale architecture, and the training technique and dataset, we conduct ablation experiments detailed below.

**Alternate objective functions.** We compare the IXYZ objective to two other information theoretic objectives: *InfoNCE* and *Single variable InfoNCE* (Oord et al., 2018; Poole et al., 2019). In-

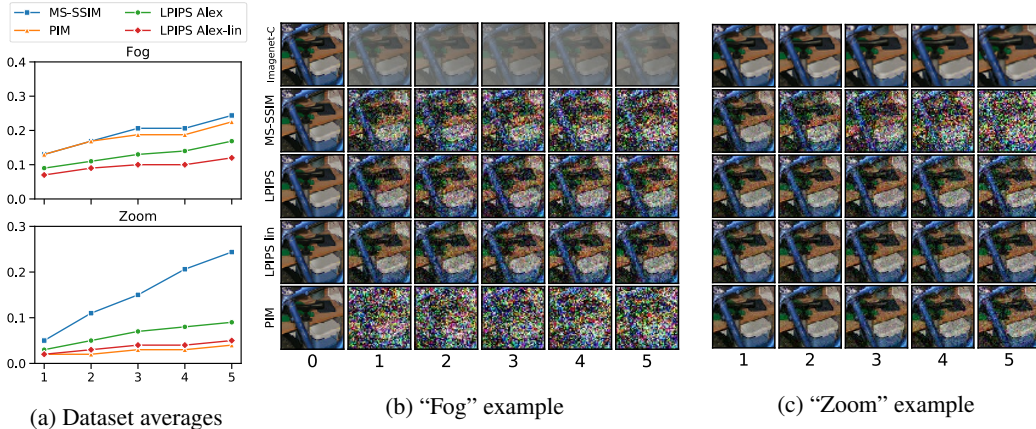

(a) Dataset averages  (b) "Fog" example  (c) "Zoom" example

Figure 3: (a): standard deviation of Gaussian noise added to images on average to yield the same metric value as the given corruption strength. (b) and (c): Each column shows equivalent amount of Gaussian noise to the corruption in the first row, according to the metric.

| CNN | 2AFC | JND |
|---|---|---|
| 4-layer | 64.88 | 51.69 |
| AlexNet | 56.8 | 37.84 |
| VGG16 | 57.31 | 36.43 |
| AlexNet (+ hidden layers) | 61.09 | 44.52 |
| VGG16 (+ hidden layers) | 67.46 | 60.07 |

| Pyramid | CNN | 2AFC | JND |
|---|---|---|---|
| Laplacian | 2-layer | 68.26 | 62.43 |
| Laplacian | 3-layer | 67.50 | 62.77 |
| Laplacian | 4-layer | 67.43 | 61.85 |
| Steerable | 2-layer | 69.00 | 61.18 |
| Steerable | 3-layer | 69.21 | 63.79 |
| **Steerable** | **4-layer** | **69.09** | **64.29** |

Figure 4: Left: BAPPS performance for various CNN architectures without using a frontend multi-scale decomposition. "+ hidden layers" indicates that the hidden layer activations were used as part of the representation (in place of the multi-scale tensors of $f_{\theta}(x)$). Right: BAPPS performance of our approach with alternate architectural choices. Bold indicates the choice used in PIM.

foNCE is a lower bound on $I(X;Y)$ that uses the same form of minibatch marginal as IXYZ. We can write InfoNCE as $I(X;Y) \geq I(Y;Z) \geq \mathbb{E}_{x,y,z\sim p(x,y)q_{\theta,\phi}(z|x)} \log \frac{q_{\theta,\phi}(z|y)}{\frac{1}{K}\sum_{i=1}^{K} q_{\theta,\phi}(z|y_i)}$. Single variable InfoNCE[4] also uses a minibatch marginal and can be written as $I(X;Z) \geq \mathbb{E}_{x,z\sim p(x)q_{\theta,\phi}(z|x)} \log \frac{q_{\theta,\phi}(z|x)}{\frac{1}{K}\sum_{i=1}^{K} q_{\theta,\phi}(z|x_i)}$. The primary difference between InfoNCE and IXYZ is that InfoNCE is not explicitly compressive. The learned representation, $Z$, is not constrained to only capture information shared between both $X$ and $Y$. Single variable InfoNCE is even less constrained, since $Z$ just needs to learn to distinguish between images ($X$) that are independently sampled in the minibatch, whereas InfoNCE and IXYZ have to be able to use $Z$ to select the correct $Y$ for the observed $X$. Training the PIM architecture using InfoNCE gives 68.99% and 64.12% on 2AFC and JND respectively, whereas single variable InfoNCE gives 65.15% and 44.79%. This shows that losing the compression and slow feature aspects of the IXYZ objective result in weaker performance for the PIM architecture.

**No multi-scale decomposition.** We trained a number of models analogous to PIM, but without enforcing approximate scale equivariance: omitting the frontend multiscale pyramid, and with only one CNN, either using only its outputs or its outputs as well as hidden layer activations in place of $f_{\theta}(x)$. For this we chose two networks used in LPIPS, VGG16 (Simonyan and Zisserman, 2015) and AlexNet (Krizhevsky et al., 2012), and the 4-layer CNN used in the frontend of PIM. Figure 4 (left) shows that none of these experiments achieved a performance close to PIM. Imposing scale equivariance thus leads to better predictions. We also note that VGG and AlexNet perform worse than the 4-layer CNN, which has much fewer parameters.

**Alternate multi-scale/CNN architectures.** We considered the Laplacian pyramid (Burt and Adelson, 1983) as an alternative linear multi-scale decomposition, and multiple alternative CNN architectures

for the frontend. Specifically, we compare against shallower variations of the 4-layer deep convolutional network that PIM uses. The results in Fig. 4 (right) show that of all these choices, the PIM architecture gives the best results, but nonetheless the others still do reasonably well. The approach thus is robust with respect to architectural details.

# 4 Related work

Early IQA metrics were based strictly on few hand-tunable parameters and architectural inductive biases informed by observations in visual psychology and/or computational models of early sensory neurons (e.g., Watson, 1993; Teo and Heeger, 1994). SSIM and its variants, perhaps the most popular descendant of this class (Wang, Bovik, et al., 2004; Wang, Simoncelli, and Bovik, 2003), define a quality index based on luminance, structure and contrast changes as multiplicative factors. FSIM (L. Zhang et al., 2011) weights edge distortions by a bottom-up saliency measure. PSNR-HVS and variant metrics explicitly model contrast sensitivity and frequency masking (Egiazarian et al., 2006; Ponomarenko et al., 2007).

Another member of this class of metrics, the Normalized Laplacian Pyramid Distance (NLPD; Laparra et al., 2016) has more similarities to our approach, in that an architectural bias – a multi-scale pyramid with divisive normalization – is imposed, and the parameters of the model (<100, much fewer than PIM) are fitted unsupervisedly to a dataset of natural images. However, the authors use a spatially predictive loss function, which is not explicitly information theoretic. While the model is demonstrated to reduce redundancy in the representation (i.e. implements efficient coding), it does not use temporal learning.

More recently, deep representations emerged from the style transfer literature, where pre-trained classifier features are used as an image embedding (Gatys et al., 2016). R. Zhang et al. (2018) and Chinen et al. (2018) trained models using IQA-specific datasets on such VGG-based representations. Ding et al. (2020) refined this by computing an SSIM-like measure on a VGG representation. As discussed earlier, these representations require collecting human responses, as well as pre-training on other tasks such as classification, that are otherwise unrelated to IQA.

A different class of IQA tasks are represented by *no-reference* quality metrics. In this paradigm, the subject is asked to rate image quality without referring to a reference (undistorted) image. Thus, no-reference metrics predict the perceived realism of an image. Typically, these metrics are targeted at detecting specific types of artefacts, such as from blurring, sensor noise, low-light conditions, JPEG compression (e.g., Ying et al., 2020), and as such do not need to generalize to arbitrary types of distortions. Thus, they are not necessarily useful as an optimization target for novel image processing algorithms, in particular if the latter are based on ANNs. To our knowledge, PIM is the first IQA metric that explicitly uses the slowness principle as an inductive bias, or that uses a contrastive learning objective.

# 5 Conclusion

In this paper, we demonstrate that making accurate predictions of human image quality judgements does not require a model that is supervisedly trained. Our model is competitive with or exceeds the performance of LPIPS, a recent supervised model (R. Zhang et al., 2018), while only relying on a few essential ingredients: a dataset of natural videos; a compressive objective function employed to extract temporally persistent information from the data; a distributional parameterization that is flexible enough to express uncertainty; a computational architecture that imposes spatial scale equivariance, as well as spatial and temporal translation equivariance. We demonstrate that our basic approach is not overly dependent on the implementation details of the computational architecture, such as the multi-scale transform or the precise neural network architecture. Instead, it is robust with respect to changes in the architecture that do not fundamentally alter the ingredients mentioned above. Not only does our method achieve a level of agreement to human ratings that matches models using human-collected data for training (e.g., classification labels and human quality ratings, such as used in LPIPS), it also gets remarkably close to matching the agreement of human ratings among each other (69.1% vs. 73.9% for BAPPS-2AFC). As such, we can interpret our results as empirical evidence that the inductive biases we employ are useful descriptions of the human visual system itself, particularly the efficient coding and slowness principles.

## Broader Impact

Many deep perceptual image metrics rely on the collection of large datasets of rated images as training data; such data could reflect the biases of human raters. While we cannot claim that PIM is free of bias, by being completely unsupervised, one possible source of bias is removed.

The broad concern about AI reducing work opportunities, in this case, seems inapplicable. Perceptual metrics are most often used as loss functions in the development of some other product, but the final quality ultimately needs to be verified by human eye. We envision this need for verification continuing into the foreseeable future.

On the other hand, good perceptual metrics have the potential to enable other research and technology which improves the lives of AI consumers, as well as reduce the burden on researchers of frequent, tiresome – and often infeasible at scale – human evaluations.

## Acknowledgments and Disclosure of Funding

No outside funding was used in the preparation of this work.

## Footnotes

[1]Code available at `https://github.com/google-research/perceptual-quality`.

[2]See Fischer (2019a) and Poole et al. (2019) for detailed descriptions of the minibatch marginal distribution.

[3]In contrast, VIB uses the same encoder for both training and inference, which means that the encoder distribution cannot be a discrete mixture, since sampling the distribution during training would require taking gradients through the discrete sample.

[4]This bound is called "InfoNCE with a tractable conditional" in Poole et al. (2019).

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
