[Supplementary Material]

# A Appendix

## A.1 Training dataset pre-processing

We used $40\,000$ publicly available videos from YouTube which were available in a spatial resolution of at least $1920 \times 1080$ pixels. In an attempt not to skew the distribution of content too far from what may inform biological representation learning, we excluded most artificial content such as screenshots and videos of computer games. We decompressed one segment of 30 consecutive frames (corresponding to 1 second) out of each video, yielding a total of ca. 11 hours of training video. To reduce video compression artifacts and prevent systematic downsampling artifacts, each segment was then spatially downsampled to a randomized height between 128 and 160. Each segment was then separated into 15 pairs of neighboring frames, and a randomly placed, but spatially colocated patch of $64 \times 64$ pixels was cropped out of each frame pair. The order of the frame pairs was then randomized in a running buffer, and all RGB pixel values were normalized to the range between 0 and 1 before being fed into the model.

## A.2 Optimization details

We trained the model on a single GPU with a batch size of 50 for $100\,000$ steps, which takes about 20 hours. All models for the ablation experiments were trained for $60\,000$ steps. We used the Adam (Kingma and Ba, 2015) optimizer with an initial learning rate of $10^{-3}$, dropping to $10^{-4}$ after $50\,000$ and to $10^{-5}$ after $80\,000$ steps, respectively. During training, we need to evaluate $\hat{p}(z)$, which is $O(MNB^2)$ in memory for a batch size of $B$ and a spatial tensor shape of $M \times N$. This can exceed the memory of a typical GPU. To resolve this problem, instead of training on the full tensors, we extract $8 \times 8$ center crops out of all tensors in $f_{\boldsymbol{\theta}}(\boldsymbol{x})$ and $f_{\boldsymbol{\theta}}(\boldsymbol{y})$. We ensure that the centers of the patches align with each other across the different scales, thus preserving translation equivariance.

To train IXYZ models, we use a Lagrange-variant proposed in Fischer (2019a):

$$\max_Z (1 - \beta)I(Z; X, Y) - \beta(I(X; Z|Y) + I(Y; Z|X))$$

$$\geq \max_Z \mathbb{E}_{x,y,z \sim p(x,y)p(z|x,y)} \log \frac{q^\beta(z|x)q^\beta(z|y)}{\hat{p}(z)p^{2\beta-1}(z|x,y)} \quad (4)$$

This technique results in simply maximizing $I(Z; X, Y)$ when $\beta = 0$, but switches to maximizing a lower bound on $I(X; Y; Z)$ when $\beta = 1$. Values of $\beta$ between 0 and 1 result in increasing compression on the $I(X; Z|Y)$ and $I(Y; Z|X)$ terms. We smoothly anneal $\beta$ from 0 to 1 over the first $10\,000$ gradient steps to avoid training trivial models where $Z$ is independent of $X$ and $Y$.

Following the methodology in R. Zhang et al. (2018), we did not select our final model based on a separate validation set, but simply by comparing results on the test set. Given our careful ablation experiments, and the fact that our model is only specified using inductive bias and is trained without supervision, we believe that the risk of overfitting to the test set is minimal.

## A.3 Details and further results on ImageNet-C experiments

To find an equivalent level of Gaussian noise which produces a metric value corresponding to a given corruption, we simply added Gaussian noise with a range of levels to the uncorrupted images in the dataset (40 distinct standard deviations ranging from 0.01 to 0.60, for image pixel values between 0 and 1), and evaluated the average metric value on the dataset. We then accepted the Gaussian standard deviation as equivalent which achieved the nearest metric value on the dataset.

This process is illustrated in Fig. 5. We plot the empirical distributions of the averaged metric values for the 5 severity levels of the corruptions, as well as the averaged metric values for a subset of levels of additive Gaussian noise. The equivalent Gaussian noise levels for all of the corruptions in the dataset are plotted in Fig. 6.

Comparing the LPIPS Alex model with LPIPS Alex-lin, the latter of which has an additional linear layer trained supervisedly on human quality ratings, we note that the additional layer appears to increase relative sensitivity to independent noise and/or decrease sensitivity with respect to other types of distortions. This is evident from the Alex-lin curve being below the Alex curve in Fig. 6 on all distortions except the top three, and the corresponding histograms being shifted to the left in

Figure 5: Histograms of metric values on the five different strengths of ImageNet-C corruptions (red) and a subset of Gaussian noise strengths (purple-green). The first row shows that the ImageNet-C Gaussian noise corruptions (with standard deviations of 0.08, 0.12, 0.18, 0.26, 0.38) exactly aligns with our additive Gaussian noise for all four models, indicating that the evaluation was correctly calibrated. The remaining rows reflect the same data visible in Figure 6 for these four selected corruptions, but giving additional information about the variance and skew of the empirical distributions of metric values. The abscissa reflects the scale of the corresponding metric; e.g., MS-SSIM ranges from 0 to 1, where 1 means the two images are identical, and 0 means the two images are maximally different. The other three models are Euclidean distances (i.e., 0 means that the images are identical). For MS-SSIM, the Shift transformation causes many images to saturate the score to 0, even with just a two pixel shift, making the empirical distributions bimodal. In contrast, the other three models are quite insensitive to the shifts and have more well-behaved empirical distributions.

Fig. 5.

More image examples are provided in Figs. 7 to 9. Fig. 7 provides more examples for the Fog and Zoom corruptions discussed in Section 3.3. The other two figures show the entire set of corruptions on the image shown in Fig. 3(b) and (c).

Figure 6: Same as Figure 3(a), but for all 15 ImageNet-C distortions, as well as our Shift and Zoom distortions.

Figure 7: More examples of the Fog and Zoom corruptions discussed in Fig. 3.

Figure 8: Visualization of all ImageNet-C corruptions with equivalent Gaussian strengths for the image in Fig. 3(b) and (c).

Figure 9: Visualization of our additional (Shift and Zoom) corruptions with equivalent Gaussian strengths for the image in Fig. 3(b) and (c).

Figure 10: CLIC 2020 human rankings according to ELO, plotted against ranking according to various perceptual quality metrics. Each dot represents one image compression method competing in CLIC 2020.