[Reviews · NeurIPS 2020]

Review 1

Summary and Contributions: The authors propose an advanced perceptual quality metric, which is learned from adjacent video frames in an unsupervised manner. This learning scheme is well motivated by the observation of human visual system. Experiments on BAPPS and ImageNet-C datasets demonstrate the effectiveness of method.

Strengths: - The proposed method is well motivated and backed up by solid theories. - Experiments are comprehensive and the overall performance is promising. - Paper is well organized and easy to follow.

Weaknesses: A good quality metric should stand the test of time. Yet it seems that the authors have not preparation to make this project publicly available. Therefore, I encourage the authors to provide the code or executable files to ensure the reproducibility.

Correctness: Technically sound, but not be carefully checked.

Clarity: Yes.

Relation to Prior Work: Yes.

Reproducibility: No

Additional Feedback:


Review 2

Summary and Contributions: An unsupervised Information-Theoretic based image quality metric using deep learning is proposed. Some experiments were conducted and showed the competitive performance.

Strengths: NOT CLEAR. Basically, I would not find the strengths of the work. Nothing could be got when reading the abstract several times. How this work is inspired by the physiology of the human visual system is not clear in the abstract. I do not think that using such words "our model is informed by the physiology of the human visual system" in the abstract means this work has new contributions. The key is to how to model the human visual system in your work. Unfortunately, this paper didn't describe it in details. There are too many DL based IQAs. What's your new contribution ?

Weaknesses: This paper is not easy to follow. It is difficult to judge the novelty. Author claimed that their model is inspired by visual physiology such as efficient coding and slowness. However, there are no any introductions about them in details. Moreover, I didn't see where two biological mechanisms are imitated in the model. I only see a DL-based IQA method without clear contributions. Finally, there are no scientific contributions in this manuscript such as " perceptual similarity is an emergent property shared across deep visual representations" discovered in [35]. [35] Richard Zhang et al. “The Unreasonable Effectiveness of Deep Features as a Perceptual Metric”. In: (2018). cite arxiv:1801.03924Comment: Code and data available https://www.github.com/richzhang/PerceptualSimilarity. URL: http://arxiv.org/abs/1801.03924. after response: Although some confusion may have been clarified, author didn't positively reply the key comments that their model is inspired by visual physiology such as efficient coding and slowness. However, there are no any introductions about them in details. "Moreover, where two biological mechanisms in details are imitated in the model is not clear". Furthermore, the paper is lacking of scientific contributions considering that it is only built on previous work.

Correctness: No

Clarity: This paper is very difficult to follow.

Relation to Prior Work: Maybe

Reproducibility: No

Additional Feedback: 1. How this work is inspired by the physiology of the human visual system is not clear in the paper. I do not think that using such words "our model is informed by the physiology of the human visual system" in the abstract means this work has new contributions. The key is to how to model the human visual system in your work in details in order to improve the current IQA. Unfortunately, this paper didn't belong this one. 2. Refer to [35] for improvement such as enhancing the scientific contributions. [35] Richard Zhang et al. “The Unreasonable Effectiveness of Deep Features as a Perceptual Metric”. In: (2018). cite arxiv:1801.03924Comment: Code and data available https://www.github.com/richzhang/PerceptualSimilarity. URL: http://arxiv.org/abs/1801.03924.


Review 3

Summary and Contributions: This paper proposed a new unsupervised, information-based perceptual quality metric, i.e. PIM. The method is based on optimization of a lower bound of the multivariate mutual information. This proposal has roots in two prominent ideas in neuroscience, efficient coding and slowness. The authors implemented the proposal in deep neural networks, and test it on BAPPS and ImageNet-C. They reported competitive performance of this method to the supervised methods.

Strengths: I very much enjoyed reading this manuscript. The method is to my knowledge novel and principled. It is well founded in Information theory, and broadly inspired by a couple of core principles in neural information processing. The method is purely unsupervised, and does not need human psychophysical judgements to train the model. Yet the method show competitive performance with the fully supervised approach in ref [35]. I found this to be quite remarkable and potentially quite significant. The problem studied in the paper is also high relevant to the NeurIPS community. ** added after rebuttal: After reading through other reviewers' comments and the authors' feedback, I remain very positive of this paper. I think the idea in the paper is novel, the experiments were reasonable and the results are very promising.

Weaknesses: My main concern is that the results are a bit preliminary (lacking more comprehensive comparison to some of the previous methods, such as ref [29] as the authors acknowledged), although the results look very promising for sure. It is also not entirely clear whether the improvement of the performance mainlycomes from the generic advantage of the objective function, or the choice of the hyper-parameters or the model architecture.

Correctness: The claims and method are sound.

Clarity: The paper is well written. The presentation in Section 3.4 perhaps could be improved. Right now, it is a bit difficult to get the key message.

Relation to Prior Work: The relation to prior work is generally well discussed.

Reproducibility: Yes

Additional Feedback: Fig 2 and Fig3 could benefit by have a bit more detailed figure legends.


Review 4

Summary and Contributions: The paper proposes PIM (Perceptual Information Metric) which is an image quality metric learned in an unsupervised manner by enforcing two loss functions - 1. Compression and 2. Consistency across time. The authors compare PIM to other proposed metrics on multiple datasets and show improvements. They also do ablation studies to show how each of the choices made in the paper yield various improvements.

Strengths: Competitive results on multiple benchmarks, ablation studies, additional qualitative experiments on ImageNet-C

Weaknesses: The paper is built on previous works and one of the main new directions proposed is the notion of consistency (perceptual metric not changing between immediate or nearby frames in a video). While this intuition seems reasonable I worry that many artifacts like - motion blur, face blur, aliasing etc. are things that could change perceptual metric significantly. It would be great to hear from authors on whether they worked on uncompressed video and if not talk a little more about the importance of using consistency metric. Since, this is one of the main novelty of the paper I would like to make sure this part is well justified.

Correctness: Yes, I did not see anything wrong in the setup and the claims made by the authors.

Clarity: Yes, for most part. The results section can be improved further, especially maybe contrast where PIM performs better qualitatively and talk about why.

Relation to Prior Work: Yes, a latest reference that might be related is shared below: From patches to pictures (PaQ-2-PiQ): Mapping the perceptual space of picture quality Z Ying, H Niu, P Gupta, D Mahajan, D Ghadiyaram, A Bovik

Reproducibility: Yes

Additional Feedback:

[Author Response · NeurIPS 2020]

We thank the reviewers for their insightful comments. We address individual notes below.

**Reviewer 1, Q3:** We agree that the utility of a perceptual metric is much increased if it is made available for public
scrutiny and further research. We are committed to making the code and model parameters available online upon
publication.

**Reviewer 1, Q7:** We have done our best to describe the details of our experiments in the paper and the supplemental
material. Due to licensing restrictions, we may not be able to make the training set public, but we will make the code
and trained model parameters available to complement the description.

**Reviewer 2, Q2:** We thank you for your honest comments. Apparently there is some conceptual confusion here;
apologies if this is due to our writing. The novelty of our approach is that we do not rely on supervised training data to
solve the task (IQA), which we mention in the title, the abstract, and many points throughout the paper. Instead, we
build a representation solely based on combining known physiological properties of the human visual system (scale
and shift-equivariance; cell responses in cortical area V1, for instance, have been shown to have these properties) with
two theories of the visual system's high-level computational goals: slowness and efficient coding. As we describe in
Sec. 2.1 and 2.3, the efficient coding constraint is explicitly implemented by the compressivity of the IXYZ objective;
the slowness principle is implemented by extracting the mutual information of successive frames from the training
video dataset, using the objective. The fact that our learned representation is highly predictive of human quality
judgements (69.4% accuracy compared to 73.9% agreement from human to human on the 2AFC task [35]) not only
represents a contribution in the engineering sense (as a way to reduce costly gathering of human responses compared to
supervised learning), but can also be interpreted as evidence that confirms the validity of the slowness and efficient
coding principles. We have not focused on the latter point, but we will certainly improve our paper in that regard.

**Reviewer 2, Q8:** We are of course aware of [35], as we compare to their empirical results directly in Table 1. Although
qualitatively comparing IQA metrics is difficult, we make some interesting observations about their approach vs. ours in
Section 3.3. In particular, their method appears to inherit certain representational properties of a classifier, like relative
equivariance to substantial amounts of fog, as it is built on pre-trained classifiers, while ours does not. Of course,
humans also are sensitive to the presence of fog and many other differences that classifiers are often explicitly trained to
ignore.

**Reviewer 2, Q11:** We did discuss the broader impact of our work in a dedicated section, as required by the submission
guidelines.

**Reviewer 3, Q3 & Reviewer 4, Q5:** We agree that more analysis of the learned representation would be helpful, and
have already begun preparations to carry out further experiments. We are also interested in evaluating whether our
representation could be useful for other tasks than IQA. However, we also think that 69.4% accuracy on the 2AFC task,
notably without any additional supervised regression stage, is an extremely strong result compared to 73.9% agreement
from human to human [35] (which we will stress in the revised paper). In our view, this result justifies publication.

**Reviewer 3, Q3/Q5:** Please note that Section 3.4 is entirely devoted to assessing which of the inductive biases
contribute to achieving good prediction results. Notably, a multi-scale architecture, or at least reading off activations at
multiple layers, appears to be crucial; compressiveness of the objective appears to improve results substantially. Other
architectural choices don't appear to matter as much. Some desirable ablation experiments are infeasible, e.g., dropping
the convolutionality constraint would lead to a many times larger model, making it difficult to train in practice.

**Reviewer 3, Q8:** We agree that Figure 3 in particular can have a more descriptive caption, and we will improve this.

**Reviewer 4, Q3:** First, let us clarify that the slowness principle, as implemented in our paper, does not necessarily
imply temporal persistence of the metric, it implies temporal persistence of the representation in a probabilistic sense
(i.e., on average, neighboring frames will have a small perceptual distance, but individual frames may have a large
distance under the metric). An intuitive depiction of the concept can be found in `http://www.scholarpedia.org/`
`article/Slow_feature_analysis`, Fig. 2. It is correct that to train our model, we would ideally use uncompressed
video, since otherwise our representation may become insensitive to compression artifacts. In practice, uncompressed
video with sufficient variability is difficult to come by. We therefore applied pre-processing to our training set, including
downsampling by a large randomized factor, in order to minimize compression artifacts, as described in the appendix.
Spot checking the sensitivity of the metric to various corruptions in Section 3.3 and the appendix, we find that it does
not have any obvious defects that may stem from compressed training data.

**Reviewer 4, Q6:** Thank you for this reference, which we weren't aware of. We will discuss this paper in the revision.
While this work uses deep learning as well, it is an example of a supervisedly trained no-reference metric, which is
targeted specifically at predicting quality of images "in the wild" without comparing to an original. Our work presents
a full-reference metric and aims more at the question of which inductive biases can help unsupervised learning, in
particular to make training perceptual metrics more data efficient.

[Meta-Review · NeurIPS 2020]

This paper proposes a novel perceptual image quality evaluation metric based on unsupervised method that aims optimization of a lower bound of the multivariate mutual information. The method is implemented using deep neural networks and tested two datasets, BAPPS and ImageNEt-C and was shown to achieve results comparable with supervised methods. The approach is well-motivated and clearly presented, and tackles an important problem without requiring subjective human judgements. One of the reviewers raise the question of applicability of the approach on compressed images. Others also suggest the experimentation part of the paper is somewhat preliminary.